# Effects of Temperature on Transparent Exopolymer Particle Production and Organic Carbon Allocation of Four Marine Phytoplankton Species

**DOI:** 10.3390/biology11071056

**Published:** 2022-07-14

**Authors:** Kangli Guo, Jie Chen, Jian Yuan, Xiaodong Wang, Shuaishuai Xu, Shengwei Hou, Yan Wang

**Affiliations:** 1Fourth Institute of Oceanography, Ministry of Natural Resources, Beihai 536007, China; guokl@mail.sustech.edu.cn; 2College of Life Science and Technology, Jinan University, Guangzhou 510632, China; pouchetii@gmail.com (X.W.); ssxu0922@163.com (S.X.); 3Department of Ocean Science and Engineering, Southern University of Science and Technology, Shenzhen 518055, China; housw@sustech.edu.cn; 4Key Laboratory of Tropical Marine Ecosystem and Bioresource, Fourth Institute of Oceanography, Ministry of Natural Resources, Beihai 536007, China; 5Department of Veterinary Diagnostic and Production Animal Medicine, Iowa State University, Ames, IA 50011, USA; leon19841@gmail.com; 6State Key Laboratory for Marine Environmental Science, Institute of Marine Microbes and Ecospheres, Xiamen University, Xiamen 361005, China

**Keywords:** transparent exopolymer particles, dissolved organic carbon, particulate organic carbon, temperature, phytoplankton

## Abstract

**Simple Summary:**

TEP is the bridge mediating DOC and POC conversion in the ocean, which is a key component of the marine carbon cycle. TEP accelerates the accumulation or deposition rate of organic carbon at the sea surface and promotes the short- and long-term sequestration of carbon, counteracting atmospheric CO_2_ increase and global warming. Temperature, as the main driving factor of carbon fixation during phytoplankton photosynthesis, is closely related to TEP production. However, little is known about the effects of temperature on TEP the production, DOC secretion, and carbon pool allocation of phytoplankton. This study analyzed the effect of temperature on the carbon pool allocation of phytoplankton and the significance of TEP in the marine carbon pool. Our results suggest that increased temperature affects carbon pool allocation in phytoplankton cells by promoting DOC exudation and extracellular TEP formation. This study provides an important basis for understanding the contribution of TEP to the allocation of POC and DOC and will benefit the prediction of phytoplankton TEP production and the marine carbon cycle under the background of global warming.

**Abstract:**

Transparent exopolymer particles (TEP) are sticky polymeric substances that are commonly found in the periphery of microbial cells or colonies. They can naturally flocculate smaller suspended particles into larger aggregates and thus play a crucial role in the biological pump and the global carbon cycle. Phytoplankton are the major contributors to marine TEP production, whereas the way TEP production interacts with abiotic factors at the species level is generally unknown but critical for estimating carbon fluxes. In this study, the effects of temperature on TEP production and carbon allocation were studied in two representative diatom species (*Nitzschia closterium* and *Chaetoceros affinis*) and two model dinoflagellate species (*Prorocentrum micans* and *Scrippisella trichoidea*). The results showed that temperature had a significant impact on TEP production in all species. First, increased temperature promoted the TEP production of all four species. Second, elevated temperature affected the carbon pool allocation, with enhanced dissolved organic carbon (DOC) exudation in the form of TEP in all species. The TEP-C/DOC percentages of *N. closterium* and *P. micans* were 93.42 ± 5.88% and 82.03 ± 21.36% at the highest temperature (24 °C), respectively, which was approximately two to five times higher than those percentages at 16 °C. In contrast, TEP’s contribution to the POC pool is lower than that to the DOC pool, ranging from 6.74 ± 0.79% to 28.31 ± 1.79% for all species. Moreover, phytoplankton TEP production may be related to cellular size and physiology. The TEP content produced by the smallest *N. closterium* (218.96 ± 15.04 fg Xeq./μm^3^) was ~5 times higher compared to *P. micans*, *S. trichoidea*, or *C. affinis*. In conclusion, TEP production is temperature sensitive and species specific, which should be taken into consideration the regarding TEP-mediated oceanic carbon cycle, particularly in the context of global warming.

## 1. Introduction

Phytoplankton can exude organic matter [1] such as polysaccharides, amino acids, proteins, and lipids to the extracellular space during the growth and reproduction process [2,3] among which polysaccharides are the most abundant macromolecules, accounting for ~70–94% of extracellular organic matter [4]. Exopolysaccharides can aggregate into transparent colloidal sticky particulate matter through a coagulation or foaming process, namely transparent exopolymer particles (TEP) [5,6]. The gelatinous TEPs are formed by spontaneous condensation of dissolved precursors under the influence of turbulence, laminar flow, ingestion, and other processes [7,8]. They can further condense with other soluble substances and detritus to form larger organic particles and precipitate as particulate organic matter (POM). Thus, TEPs play an important role in the size conversion from the dissolved organic carbon (DOC) to the particulate organic carbon (POC) [9,10,11]. Therefore, a better understanding of the carbon contributed by TEP in DOC and POC is crucial in order to investigate the fates of organic matter in the ocean [7,12].

Phytoplankton blooms are the main source of marine TEP production [6,13,14]. In recent years, the frequency and duration of harmful algal blooms have increased in the South China Sea, leading to massive accumulation of TEP in spring. These blooms were dominated by several dominant diatom (*Nitzschia closterium* and *Chaetoceros affinis*) or dinoflagellate (*Prorocentrum micans* and *Scrippisella trichoidea*) species in the South China Sea [15,16]. Sedimentation of phytoplankton cells with TEP is a major component of sinking carbon during the late stage of the blooms, although a considerable fraction of the carbon could be consumed by respiration of heterotrophic microbes during the sinking [17]. Therefore, it is necessary to understand how TEP production and carbon fluxes are affected by environmental factors better using dominating diatoms and dinoflagellates as representatives. Fukao et al. [18] discovered that TEP production of phytoplankton may be specific to a few species. In addition, because diatoms and dinoflagellates belong to two different taxonomic units in the Infrakingdom rank, they might have distinct physiological responses and should be studied separately [19]. Temperature is a key factor affecting photosynthesis and phytoplankton growth [20,21,22]. Previous studies proved that diatom and dinoflagellate blooms mainly occurred in the South China Sea where the water temperature could rise from 15 °C in winter to 27 °C in spring [15,16]. This implies that the dramatic temperature change between winter and spring might have an essential role in shaping phytoplankton growth and bloom formation in this region. It has been reported that temperature changes can affect the enzymatic activities and metabolic rates of different phytoplankton. Hence, it might also affect carbon excretion [23,24,25] and TEP production and aggregation [26,27,28]. However, the relationship between cellular carbon pool allocation and TEP production in phytoplankton is still not clear. Given the important role of TEP in connecting DOC and POC, it is essential to understand how increasing temperature could affects the TEP production from DOC and POC, which is of great significance to analyze the role of TEP in the ocean carbon cycle under the global warming scenarios. Here, we cultivated two diatom and two dinoflagellate species isolated from the South China Sea at three temperatures (16 °C, 20 °C, and 24 °C) in order to evaluate the carbon pool allocation between DOC and POC, and the influence on TEP production. We hypothesize that the contribution of DOC and POC to TEP production is independent of temperature.

## 2. Materials and Methods

### 2.1. Culture Condition

The two diatom species (*Nitzschia closterium* and *Chaetoceros affinis*) and the two dinoflagellate species (*Scrippsiella trichoidea* and *Prorocentrum micans*) used in this study are the dominant species of coastal phytoplankton blooms in the northern regions of the South China Sea (Table 1). The four phytoplankton species were cultured at three temperatures (16 °C, 20 °C and 24 °C) for two weeks till the exponential phase and were then inoculated into conical flasks and re-cultured in media prepared from 0.7 μm of membrane-filtered artificial seawater with salinity of 30 ± 1 psu. The cultures were then placed in an illuminated incubator with a light intensity of 100 μmol photons/(m^2^·s^2^) in a 12 h:12 h illumination–darkness cycle. To ensure axenic cultivation, penicillin and streptomycin sulfate were added to final concentrations of 0.04 μg/mL and 0.01 μg/mL, respectively.

### 2.2. Measurement of Phytoplankton Abundance and Biomass

Aliquots of 5 mL cultures were taken every day and preserved with 0.5 mL of acidic Lugol’s iodine solution. The cells were counted, and cell dimensions were determined using an inverted microscope (OLYMPUS-CKX41). When calculating the cell volumes, we assumed that *C. affinis* was cylindrical, *N. closterium* was double cone shaped, *S. Trichoidea* was spherical, and *P. micans* was ellipsoid. The specific growth rates at different temperatures were calculated using Formula (1) below [29].
(1)$$\mu =\frac{ln\;\left(N_{1}\right)-ln\left(N_{0}\right)}{\left(t_{1}-t_{0}\right)}$$

Here, *μ* is the specific growth rate; *N*_1_ and *N*_0_ are the numbers of cells at the end and beginning of the exponential phase, respectively; and *t*_0_ and *t*_1_ represent the beginning and end time of the exponential phase, respectively.

### 2.3. Determination of Chlorophyll a (Chl a) Concentrations

Chl *a* were determined by filtering 15 mL of culture through a 0.7 µm glass fiber membrane. Chl *a* samples were extracted in the darkness in 90% acetone solution for 24 h, and fluorescence of extracted Chl *a* was measured with a UV–vis spectrophotometer.

### 2.4. Colorimetric Determination of TEP

During the stationary phase, 3 mL of culture was taken from each species and filtered through a 0.4 μm pore-sized polycarbonate membrane with a negative pressure lower than 0.02 MPa. The filter membrane was stained in 1 mL of 0.02% Alcian blue (pH = 2.5, containing 0.06% acetic acid) for five seconds and then washed twice with 1 mL ultrapure water. The filter membrane was transferred to a 25 mL beaker with 6 mL of 80% sulfuric acid solution and incubated for 2 h. Finally, the absorbance value of the solution was measured by a UV–vis spectrophotometer at the wavelength of 787 nm, and the content of TEP was calculated by constructing a xanthan gum (Xeq.) standard curve [30].

### 2.5. Calculation of TEP Carbon Content

TEP carbon content (TEP-C) was calculated by the following formula:TEP-C = (0.51~0.88) × TEP_color_(2)

TEP_color_ refers to the TEP concentration measured by the colorimetric method, and 0.51–0.88 is the range of the coefficient for different phytoplankton species [31]. The range of coefficient described in this study was 0.75 for diatoms and 0.51 for dinoflagellate [10].

### 2.6. Determination of the POC and DOC Content

To prepare POC samples, 30 mL of algal culture were filtered through a combusted (450 °C, 4 h) 0.7μm glass fiber membrane. The organic matter caught on the membrane was used to measure POC, and the filtrate was used to measure DOC. Then the membrane was wrapped with tinfoil, dried at 40 °C for 48 h and stored at −20 °C. The filtrate was transferred to a 30 mL scintillation flask and stored at −20 °C. To measure the POC content, 5 mol/L hydrochloric acid was used to wash the membrane and then incubated at 40 °C to dry for 48 h. The membrane was loaded into a tin boat, using the membrane filtered with artificial seawater as a blank control. The POC content was determined by an element stable isotope ratio mass spectrometer. For DOC content measurement, samples were first thawed at room temperature, and then measured by a TOC measuring instrument using artificial seawater as the blank and potassium hydrogen phthalate as standards.

### 2.7. Statistical Data Analysis

GraphPad Prism 9.0 was used to process and plot the data, and SPSS 25.0 was used for statistical analysis. One-way ANOVA was conducted to analyze the TEP concentration and DOC concentration. Canoco 5 was used to analyze the effects of temperature on different physiological parameters of phytoplankton by principal component analysis (PCA). The significance level cutoff was set to 0.05 in this study.

## 3. Results

### 3.1. Effects of Temperature on Phytoplankton Growth

The four species grew faster at higher temperature (20 °C and 24 °C) than at low temperature (16 °C) (Figure 1a–d), but the specific growth rates of *C. affinis* and *S. trichoidea* were not significantly affected by varying temperatures (*p* > 0.05) (Figure 1e, Appendix A). The cell densities of *S. trichoidea* and *C. affinis* at 16 °C were significantly lower than that at 20/24 °C (*p* < 0.05) (Figure 1c,d). However, the specific growth rates of *N. closterium* and *P. micans* increased significantly with the increase of temperature (*p* < 0.05) (Figure 1e, Appendix A). As a result, the two species did not grow to the stationary phase until the 15th and 21st day at 16 °C (Figure 1a,b), respectively.

### 3.2. Effect of Temperature on TEP Production

All four species could produce TEP at the three temperatures (Figure 2). TEP concentrations of the four species ranged from 1138 to 6218 μg Xeq./L, and the concentrations normalized to Chl *a* ranged from 2.15 to 13.47 μg Xeq./Chl *a*. *P. micans*, *C. affinis*, *N. Closterium,* and *S. trichoidea* had significantly promoted TEP production with the increase of temperature (*p* < 0.05) (Figure 3). However, only *P. micans*, *C. affinis*, and *N. Closterium* enhanced significantly the production when normalized to Chl *a* (*p* < 0.05), comparing to *S. trichoidea* with unchanged productions (*p* > 0.05) (Figure 3, Appendix A).

The four species had obviously different cell sizes, and *P. micans* and *N. closterium* are the largest, (15.19 ± 5.24) × 10^3^ μm^3^, and the smallest, (0.10 ± 0.02) × 10^3^ μm^3^, respectively (Table 1). Therefore, to compare their TEP production capacity, the TEP content was normalized to the cell volume accordingly (Figure 4). At 16 °C, 20 °C, and 24 °C, the normalized TEP content from *N. closterium* was significantly higher, up to 5 times, than the other three species (*p* < 0.01) (Figure 4), while *P. micans* had the lowest values.

### 3.3. Effects of Temperature on the Allocation of Organic Carbon Content in Phytoplankton

#### 3.3.1. Effects of Temperature on DOC, POC, and TOC Contents

Increased temperature mainly affected the DOC content per cell significantly (*p* < 0.05) in most species, rather than POC or TOC (*p* > 0.05) (Figure 5a–d). The DOC content of *P. micans* and *S. trichoidea* decreased significantly with the increase of temperature (*p* < 0.05) (Figure 5a,b), while *N. closterium* held it almost constant across different temperatures (*p* > 0.05) (Figure 5d), and *C. affinis* had a peak value at 20 °C (Figure 5c). This was also true for the DOC/TOC ratios, which peaked at 20 °C for *C. affinis* (Figure 5g) but decreased significantly when the temperature went up for the other species (*p* < 0.05) (Figure 5g).

#### 3.3.2. Effect of Temperature on TEP-C, TEP-C/DOC, and TEP-C/POC

After converting TEP concentrations to carbon content, the results showed that increased temperature promoted TEP-C content significantly in all species (*p* < 0.001) except *S. trichoidea* (Figure 5a,d). Increased temperature promoted TEP production which contributed significantly to the DOC fraction in all species (*p* < 0.05) (Figure 5e) and the POC fraction in all species except *S. trichoidea* (Figure 5f). The TEP-C/DOC ratios of *P. micans* and *N. closterium* reached 82.03 ± 21.36% and 93.42 ± 5.88% at 24 °C (Figure 5e), respectively, about 3–5 times of those at 16 °C. The TEP-C/DOC ratio of *S. trichoidea* at 24 °C was 42.70 ± 1.73%, the highest across different temperatures (*p* < 0.05) (Figure 5e). The TEP-C/POC ratio of the four species ranged from 6.74 ± 0.79% to 28.31 ± 1.79% (Figure 5f). The DOC/TOC fraction decreased with increasing temperature except in *C. affinis*, despite an elevated organic carbon production.

#### 3.3.3. PCA Analysis

A principal component analysis (PCA) was used to explore the influences of temperature on different physiological parameters of phytoplankton. A total of 63.4% and 25.0% of the variance of physiological variables were explained by the first and second axes, respectively (Figure 6). Temperature has a great influence on the physiological characteristics of the four species. The two dinoflagellates (*P. micans* and *S. trichoidea*) were correlated with POC, TOC, and cell volume, and POC had a strong correlation with TOC, while the two diatoms (*C. affinis* and *N. closterium*) had good correlations with the growth rate and Chl *a*. (Figure 6).

## 4. Discussion

### 4.1. Effects of Temperature on Phytoplankton Growth

Temperature is one of the important environmental factors that affect the growth and reproduction of phytoplankton [22]. Previous studies found that the temperature range around the growth of *S. trichoidea* was 17 °C–25 °C, and the optimal growth temperature was 20 °C–22 °C [32]. This was in accordance with the findings in this study that *S. trichoidea* grew better at 20 °C–24 °C. It was also reported that the temperature around the growth of *P. micans* ranged from 18 °C to 28 °C [33]. Here, the temperatures for the growth of the four species were set between 16 °C and 24 °C, and finally they grew better at 20 °C or 24 °C than at 16 °C, implying that higher temperature enhances their growth likewise. A study by Wang et al. [16] reported that high temperature in summer would lead to the outbreak of phytoplankton blooms. Therefore, it is assumed that increasing temperatures in summer may spur the rapid growth of certain phytoplankton species in the South China Sea given appropriate solar radiation and nutrient supply, resulting in harmful algal blooms [27,28].

### 4.2. Effect of Temperature on TEP Production of Phytoplankton

The TEP productions of sister strains could be considerably disparate. In this study, the TEP concentration of the *N. closterium* strain was higher than that used by Engel et al. [34], while it was lower regarding the *C. affinis* strain versus the CCMP 159 used by Passow [31] who used 15 °C and lower temperature for culturing phytoplankton. It could be summarized from the few examples above that TEP production has intra-species or even inter-strain differences at different temperatures as a result of varied responses. Furthermore, the TEP production normalized to Chl *a* of *S. trichoidea* was not affected by temperature variation, whereas it increased under higher temperatures with respect to *P. micans*, *C. affinis*, and *N. closterium*. Claquin et al. [21] found in their study that the TEP produced by seven out of the eight phytoplankton species rose with the increase of suboptimal or optimal temperature, except for the dinoflagellate species *Lepidodinium chlorophorum* in consistency with our findings. However, Claquin et al. [21] found that the production of TEP increased with the temperature until a maximum and decreased at high temperature. The three temperature (16 °C–24 °C) conditions in our study are only the suboptimal or optimal temperatures of the four phytoplankton species, and none of them are supraoptimal or high temperature (lethal temperature). Consequently, we assumed that TEP concentration may decrease when the temperature reaches supraoptimal. A wider range of temperature studies is necessary next. Guo et al. [27] found in Jiaozhou Bay that TEP concentration increased obviously when temperature ascended in summer. Studies have also shown that a significant linear correlation exists between phytoplankton TEP production and photosynthetic activity within the proper temperature range. For example, the TEP content kept increasing with temperature going up before reaching the maximum growth temperature (24.7 °C) in the diatom *Thalassiosira pseudonana* [21]. Other studies reported that increased temperature significantly affected enzyme activities and metabolic rates of phytoplankton cells and could level up the synthetic and exudation rates of organic carbon [23,35]. Therefore, the four phytoplankton species used in this study may have higher photosynthetic rates and could produce more TEP within the temperature ranges of their normal growth in warmer regions in summer. On the other hand, temperature was the only environmental factor trialed in our study, while nutrients and illumination were maintained at optimal levels. Higher temperature promoted the production of TEP, which may be attributed to higher growth rates and enhanced carbon fixation by continuous photosynthesis in our controlled experiments. In order to maintain a balanced intracellular carbon content under high temperature (POC content did not change significantly at 16 °C–24 °C), excessive carbons were secreted into the ambience from phytoplankton cells, resulting in an increase in peripheral TEP concentration. In contrast, in the open ocean where nutrients, solar radiation, and temperature vary simultaneously, TEP production becomes unpredictable, as each factor might have different influences on it [36,37,38]. Therefore, the interactive effects of multiple factors on TEP production should be further investigated.

### 4.3. Relationship between TEP Production and Phytoplankton Physiologies

The TEP content per unit cell volume was variable for the four phytoplankton species in this study. The smallest-sized *N. closterium* had the highest TEP content, about 5 times higher than the other three species. Yet, the largest-sized *P. micans*, which was ~150 times larger than the smallest *N. closterium*, only had a TEP content 1/5 of *N. closterium*. Fukao et al. [18] reported that the cell volume of *Coscinodiscus granii* (103.00 ± 6.24) × 10^3^ μm^3^ was 400 times larger than that of *Skeletonema* sp. (0.26 ± 0.05) × 10^3^ μm^3^, while the concentration of TEP per cell volume of the former (341.6 ± 56.33 fg Xeq/μm^3^) was 5 times higher than that of the latter (68.30 ± 3.28 fg Xeq/μm^3^). This was opposite to our findings and likely indicates that TEP content does not change linearly with phytoplankton cell volume. According to Wu et al. [39], the absorption of nutrients by cells from the aquatic environment is related to the specific surface area of cells, which affects the transmembrane transportation efficiency of extracellular substances and may further be associated to the capability of intracellular organic matter exudation. Therefore, it is speculated that the content of TEP in this study may be closely related to cell volume and specific surface area. In addition, TEP production in phytoplankton may be connected with the physiologies of cells. As a typical benthic diatom, *N. closterium* belongs to the class of pennatae diatoms and usually lives an adherent life to solid matrices. The dinoflagellates *P. micans*, *S. trichoidea*, and centrales diatom *C. affinis* are generally planktonic. Most diatoms have longitudinal shell fissure connecting cytoplasm and extracellular environment which can determine the attaching and gliding ability of diatom cells on a solid surface [40], but the central diatom does not have such a structure. It is confirmed by scanning electron microscopy that diatoms can secrete a large amount of polysaccharides through the shell and tube shell fissure for gliding on and attachment to the surface. Therefore, some pennatae diatoms are able to produce more polysaccharides than the central diatoms due to the existence of the longitudinal groove structure [41,42,43,44]. This may explain why the TEP content of *N. closterium* was several orders of magnitude higher than that of *C. affinis*. The gliding and attaching functions of benthic diatoms depend on the exopolysaccharides on the cell surface [44,45,46]. These exopolysaccharides aggregated together to form a strong membrane of TEPs. Previous studies have shown that increased TEP production could facilitate the growth and reproduction of benthic diatoms. TEP may participate in the benthic diatoms’ attachment to hard surfaces, helping them to endure harsh environmental conditions such as turbulence [47]. Therefore, it can be inferred that *N. closterium* produces TEP consistently to become resistant to the frequent environmental changes. In conclusion, the amount of TEP produced by phytoplankton may be associated with the species’ physiologies and growth patterns, so the relationship between TEP and physiologies should be further studied.

### 4.4. Effects of Temperature on TEP-C and Allocation of Organic Carbon Content

This study showed that the POC and TOC contents per cell of the four species did not change with temperature, in consistence with the study by Montagnes et al. [23] who concluded that temperature had no significant effect on the intracellular carbon content by studying eight phytoplankton species. However, some researchers pointed out that the change in POC content resembled a parabolic curve (concave upward) as temperature increased [48,49]. Thompson et al. [50] studied the changes to the carbon and nitrogen contents of eight phytoplankton species with varying temperature, and only three diatoms showed the parabolic shape (concave upward), while the other five species exhibited no significant changes at all, possibly caused by different temperature-induced responsive mechanisms of the carbon storage. Part of the organic carbon fixed by photosynthesis of phytoplankton is exudated as DOC, which contributes ~5–30% of the total carbon and is one of the DOC sources in the ocean [4,51,52]. Castillo et al. [53] found that the DOC/TOC percentages of different phytoplankton varied greatly, which was in agreement with the results of this study. The DOC/TOC percentages of *S. trichoidea* (34.09 ± 2.33%) and *P. micans* (16.72 ± 2.24%) ranked at the top and bottom, respectively, of which the latter lay within the reported range (~5–30%) by Baine et al. [51]. Consumption and absorption of DOC by phytoplankton have not been taken into account in this study, leading to underestimation of the actual DOC secretion. This may elucidate the great difference in DOC secretion by different species. However, it is the net DOC measured here that has an effect on other sea life or contributes to the ocean carbon pool [54]. In general, DOC secreted by phytoplankton can be divided into two categories: the compounds with specific functions (e.g., extracellular enzymes, toxins, lectins, nucleic acids, etc.) and compounds with no known functions. Thornton [54] argued that DOC secretion occurs at the expense of loss of resources (i.e., sequestration of carbon and energy), and that most DOC secretion either has negligible or unknown functions on phytoplankton. Therefore, it may be necessary to further explore the composition and latent function of DOC in the future.

DOC contents of *P. micans* and *S. trichoidea* decreased, and yet the TEP-C/DOC ratios increased as temperature increased. The DOC content per cell of *N. closterium* was not affected by temperature, and the TEP-C/DOC ratio also increased with the increase of temperature. For *C. affinis**,* the DOC content was the lowest at 24 °C, but the TEP-C/DOC ratio reached the highest value. Furthermore, the DOC/TOC fraction decreased with increasing temperature. These results indicate that higher temperature can expedite the conversion of DOC to TEP-C, causing enhanced DOC secretion in the form of TEP (Figure 7). This aligns with the results of Seebah et al. [55] that temperature promoted TEP condensation. The percentage of DOC production ending in TEP ranged from 3% to 40% in nutrient-rich and oligotrophic waters, respectively [56]. Whereas this range was much lower than what we found. For example, TEP-C in the DOC of *N. closterium* and *P. micans* were 93.42 ± 5.88% and 82.03 ± 21.36%, respectively. The reasons could be inferred as follows. First, the DOC and TEP contents were measured in the stationary phase of cultures in this study. The cell density reached the maximum, and the large DOC accumulation most likely increased the opportunities for the collision of internal particles to form more TEP. Second, the increase in temperature could promote TEP formation [55,57], so the TEP-C/DOC percentages of the four species became reasonably the biggest at the highest temperature (24 °C) in this study. Based on the importance of TEP in marine carbon flux, it was assumed that the percentage of carbon secreted during the primary production of phytoplankton is generally limited to ~10–20% [58,59]. Studies have shown that DOC contains ~23–80% carbohydrates, including ~70–94% polysaccharides, and it was assumed that DOC contains 50% extracellular polysaccharide with ~5–10% TEP precursor thereof [4]. It can thus be estimated on the basis of the theory above that the maximum amount of carbon transferred from DOC to POC driven by the TEP of the four species in this study is about half (~38.5–46.5%) of the fixed carbon in the cells, indicating that the contribution of TEP to marine carbon flux cannot be ignored. Therefore, studying the allocation of TEP between DOC and POC is of great significance to determine the contribution of TEP to the ocean carbon cycle.

We found that the TEP-C/POC ratios of the four species ranged from 6.74 ± 0.79% to 28.31 ± 1.79%. Previous laboratory studies have shown that the contribution of TEP to the POC pool ranged between 20% and 60% [56]. Some of them are higher than our findings. The percentage of TEP-C in POC in our research was lower than that in the Atlantic Ocean (~34–103%) [60], possibly due to local species richness and the structural complexity of ecosystems in the open ocean. The maximum proportion of TEP-C in POC in this study was 28.31 ± 1.79%, which is close to the reported minimum of 34% in the Atlantic Ocean. Therefore, the TEP-C contribution of phytoplankton to the marine POC pool should not be ignored given that frequent blooms occur in the coastal and open oceans. Numerous studies have shown that TEP servers as the essential substrate for marine snow and promotes diatom aggregation [7,61]. Hence, it can be speculated that the four studied phytoplankton species here could promote the deposition of microbial cells and organic substances by generating a large quantity of TEPs during the blooms given appropriate higher temperatures.

### 4.5. PCA Analysis

PCA is an effective mathematical tool used to determine the main physiological parameters on a spatial scale via dimensionality reduction [62]. It was found in this study that temperature has a positive effect on the Chl *a* and specific growth rate of diatoms. Therefore, given sufficient nutrients and appropriate illumination, rising temperature can promote diatom blooms in some marine ecosystems, such as coastal up-welling regions or estuaries. Temperature can promote the content of POC, DOC, TOC, and cell volumes of dinoflagellates, indicating that their carbon contribution to the marine carbon pool will be enhanced at higher water temperature. This is similar to the results of Xiao’s study, that is, within their growth temperature ranges, diatoms and dinoflagellates do not respond to their physiological characteristics but respond to changes in environmental conditions by adjusting their physiology [63].

## 5. Conclusions

The four phytoplankton species grew better at high temperatures in this study, and the specific growth rate was affected by temperature change with interspecies differences. The increase of temperature promoted the production of TEP in *P. micans*, *N. closterium*, and *C. affinis* rather than *S. trichoidea* when normalized to the Chl *a* concentration. The POC and TOC contents per cell of the four species did not change with temperature, but the DOC content of *P. micans* and *S. trichoidea* showed an opposite trend to the increasing temperature. The TEP-C content and the TEP-C/DOC ratio increased significantly as the temperature went up (*p* < 0.001). These results indicate that higher temperatures can promote the conversion from DOC to TEP-C, and as a result, more DOC is secreted by cells in the form of TEP. Moreover, the production of TEP in phytoplankton may be related to the cell size and physiological status. The content of TEP produced by the smallest-sized *N. closterium* was about 5 times higher than that of *P. micans*, *S. trichoidea*, and *C. affinis*. These observations overturn our hypothesis that the contribution of DOC and POC to TEP production is independent of temperature and improved our understanding of other factors affecting TEP production, which is required to make realistic projections of carbon fluxes in a warmer ocean.

## Figures and Tables

**Figure 1 biology-11-01056-f001:**
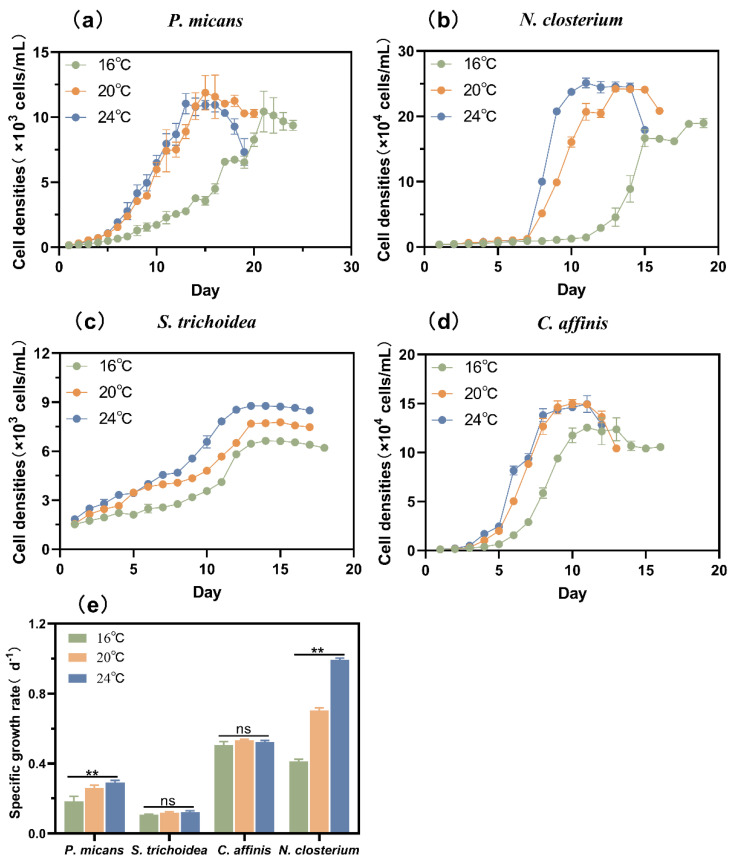
(**a**–**d**) Growth curves and (**e**) specific growth rates of the four phytoplankton species at different temperatures. **: *p* < 0.05.

**Figure 2 biology-11-01056-f002:**
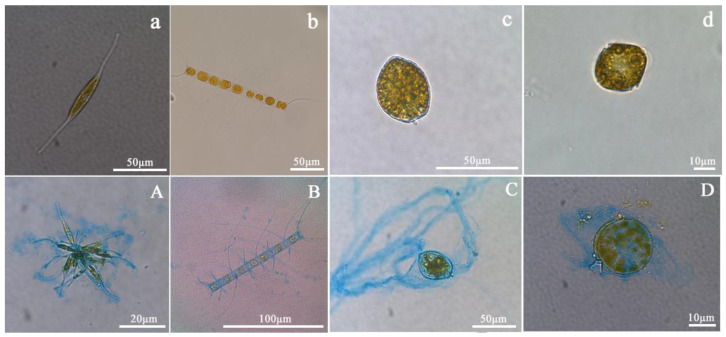
(**a**–**d**) The morphology of the four species under a light microscope. (**A**–**D**) Alcian blue staining of TEP released by the four species. (**a**,**A**) *N. Closterium*; (**b**,**B**) *C. affinis*; *(***c**,**C**) *P. micans*; (**d**,**D**) *S. trichoidea*.

**Figure 3 biology-11-01056-f003:**
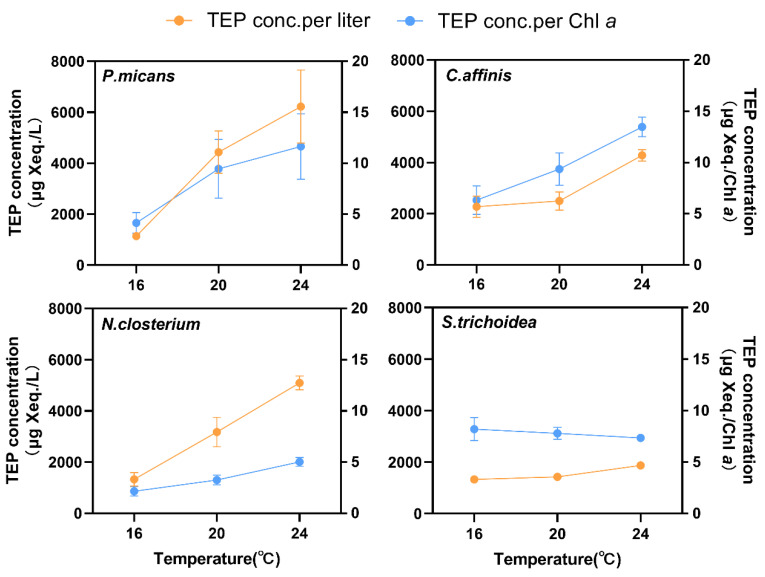
Effects of temperature on the TEP concentrations of the four species.

**Figure 4 biology-11-01056-f004:**
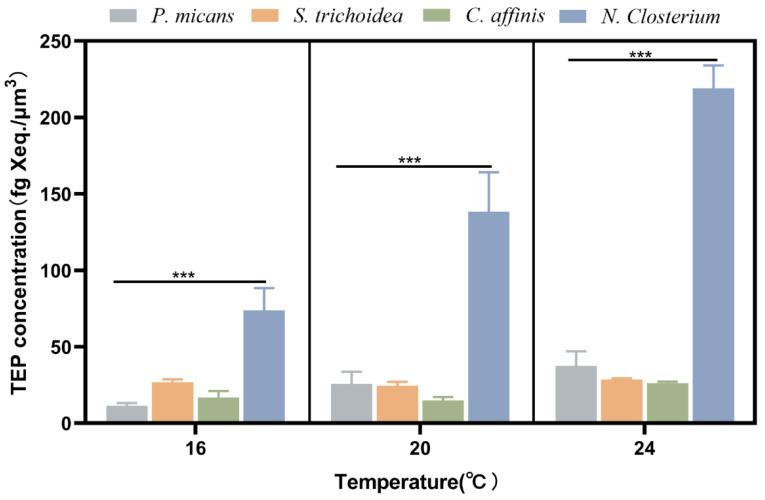
Comparison of TEP content per unit cell volume of the four species at different temperature. ***: *p* < 0.01.

**Figure 5 biology-11-01056-f005:**
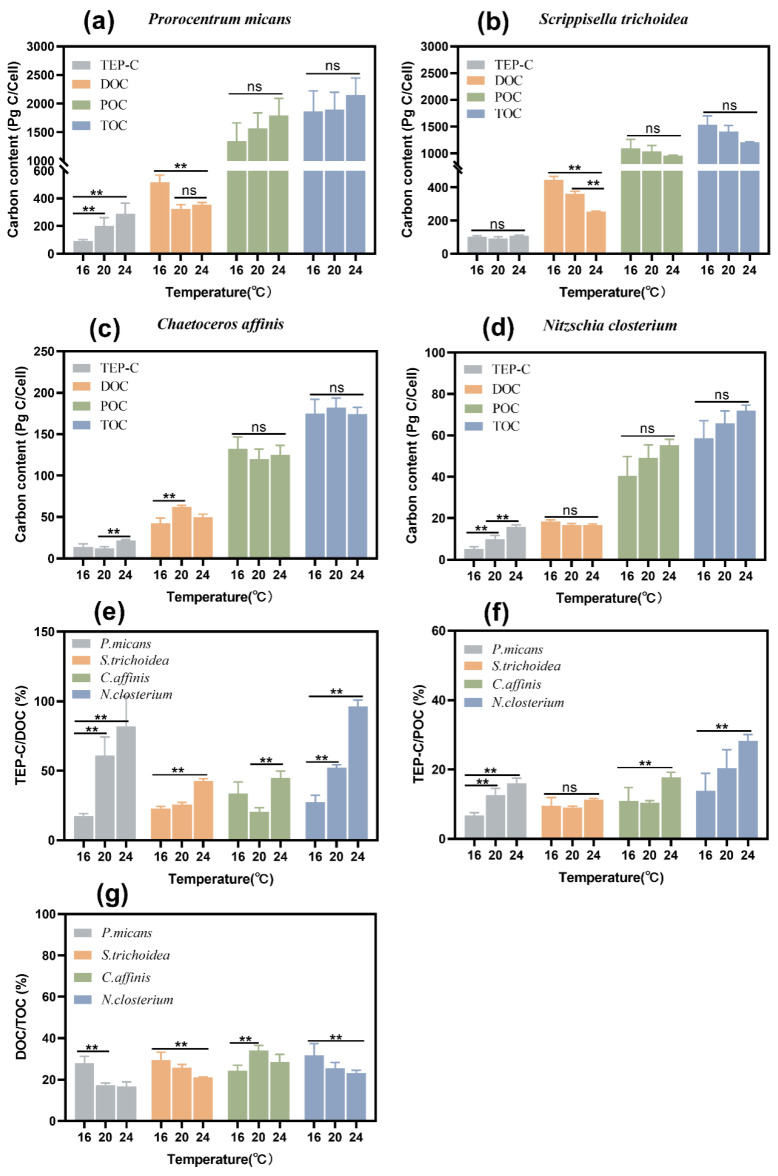
Alterations in carbon content of *P. micans* (**a**), carbon content of *S. trichoidea* (**b**), carbon content of *C. affinis* (**c**), and carbon content of *N. closterium* (**d**) at different temperatures. Alterations percentage of TEP/DOC (**e**), percentage of TEP/POC (**f**) and percentage of TEP/TOC (**g**) at different temperatures. Values are presented as mean ± SEM of three replicates (n = 3). ** *p* < 0.05 indicate significant difference.

**Figure 6 biology-11-01056-f006:**
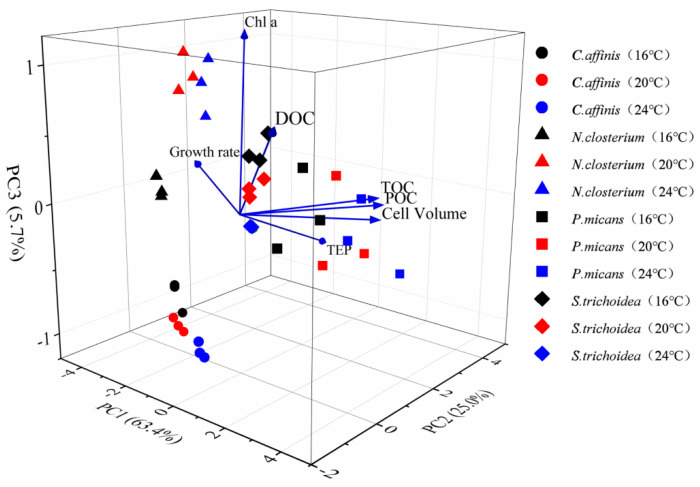
The effects of temperature on the different physiological parameters of phytoplankton by PCA.

**Figure 7 biology-11-01056-f007:**
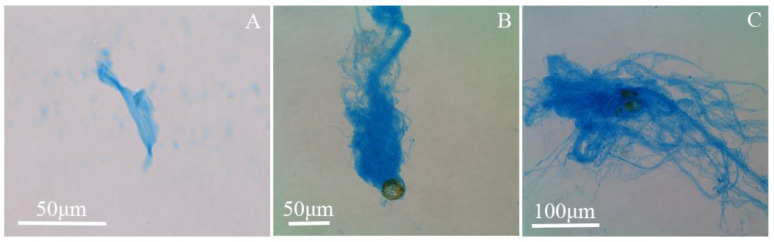
The transformation process of dissolved TEP to a granular state of *P. micans* at different temperatures. The temperatures in (**A**–**C**) were 16 °C, 20 °C, and 24 °C, respectively.

**Table 1 biology-11-01056-t001:** The equivalent spherical/cone diameters, cell volumes, and collection sites of the four phytoplankton species used in this study.

Species	Equivalent Spherical/Cone Diameter (μm)	Cell Volume (×10^3^ μm^3^)	Initial Cell Density (Cells·mL^−1^)	Collection Site
*Prorocentrum micans*	37.30 ± 3.29	15.19 ± 5.24	200	Daya Bay
*Nitzschia closterium*	22.44 ± 3.38	0.10 ± 0.02	3000	Pearl River Estuary
*Chaetoceros affinis*	15.29 ± 1.83	1.12 ± 0.41	1000	Hong Kong Tsing Yi
*Scrippisella trichoidea*	26.65 ± 4.90	7.45 ± 0.29	1000	Pearl River Estuary

## Data Availability

Not applicable.

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
