# Peer review of "Effects of Temperature on Transparent Exopolymer Particle Production and Organic Carbon Allocation of Four Marine Phytoplankton Species"

_biology, 2022, doi:10.3390/biology11071056_

Round 1
Reviewer 1 Report
Kangli Guo and colleagues present an interesting study on the temperature effect on TEP production of four common phytoplankton species. Technically, this study appears to be well executed but I am not an expert of TEP methodology and physiology. The approach is not novel; however, the results merit publication because species-specific responses of TEP production and growth rates to increased temperature in the course of global warming are needed. The manuscript needs to be improved in style and language. I made numerous comments on the pdf that may help improve the technical quality of the manuscript. Please note that I did not do exhaustive language editing. The Discussion lacks focus and should be shortened. Importantly, in particular with respect to the topic of this SI emphasizing "Multifunctionality in Response to Environmental Changes", in their conclusions the authors need to consider interactive effects of temperature, nutrients and light for phytoplankton growth rates and TEP production. The coastal area of the South China Sea is a shallow, eutrophic environment that is not representative for the ocean at a global level.

Reviewer 2 Report
I have reviewed manuscript biology-1755581 by Guo et al. Whenever I review a manuscript, the first question I ask is whether the study involved application of the scientific method. The scientific method involves asking questions and/or testing hypotheses. In this case, the answer is yes. The last sentence of the Introduction says: We hypothesis (sic) that TEP production and DOC concentration will increase with Temperature.
The problem I see with this manuscript is that the issues of TEP production and DOC production by phytoplankton have been thoroughly studied in the past. Much of the early work on TEP production was done by Passow and Alldredge between 1995 and 2002 (references 33–35), and the study of Claquin et al. (2008) specifically looked at temperature effects on TEP production. The whole issue of DOC production by phytoplankton was studied ad nauseam by Fogg in the 1960s and 1970s (e.g., Oceanogr. Mar. Biol. Ann. Rev. 4: 195–212 [1966]) and by Sharp (Limnol. Oceanogr. 22(3): 381–399 [1977]). I think it is provocative that neither author is cited in this manuscript. Maybe these papers have been forgotten because they were published ~50 years ago, but if I were going to sit down and write a paper about the effects of temperature on DOC excretion by phytoplankton, I would certainly want to look through the voluminous early literature on the subject of phytoplankton DOC excretion.
It is unclear to me that this study of four species in stationary phase tells us anything that we did not already know.
Some of the choice of words indicates a lack of familiarity with the literature. For example, the cultures were sampled in the stationary phase of growth. The stationary phase follows the log phase and is the time when the cell counts are no longer increasing. It is not referred to as the “steady status” (line 166). The four species do not “grow uncontrollably” during the summer. They may grow faster during the summer because the temperatures are higher, but their growth is not uncontrolled. A “U-shaped” curve (line 321) is concave up or convex down.
Reviewer 3 Report
The manuscript under review contains the results of an interesting study on the secretion of transparent exopolymer particles (TEC) by four species of "algae" capable of creating blooms. The title of the manuscript and the "Introduction" formulate the research problem - the influence of temperature on the secretion of TEC. However, the literature review in the "Introduction" section shows that a relatively large number of studies have been devoted to this issue, and all the cited publications indicate that such an impact has already been found before. Therefore, in principle, it is not known for what reasons this research was undertaken. Although the Authors in lines 79-80 indicate that “…. how temperature influences the carbon pool allocation in DOC and POC remains enigmatic ”.
Meanwhile, the "Introduction" does not address two important issues: (1) The experimental cultures were conducted with different temperatures, from 16 ℃ to 24 ℃ with an interval of 4 ℃. Two species of diatoms were selected for the study - "typical benthic diatom, N. closterium belongs to the class Pennatae" and C. affinis (centrales), which is generally planktonic, as well as two species of dinoflagellates - P. micans, S. trichoidea. However, this clear distinction does not appear until midway through the discussion (lines 296-298). In the "Material and methods" section (lines 95-97) and in the abstract (lines 34-35) it is only indicated that all these species are capable of forming plankton blooms.
Thus, the research covered species of significant systematic distance, i.e. belonging to two different systematic units in the Infrakingdom rank, which results in a large physiological difference of cells. Moreover, the tested "algae" differed in life form. Dinoflagelates are monads, and therefore able to move actively with the help of flagella, which feed on by the method of myzocytosis. and Diatoms are unicellular plants that are surrounded by a permanent cell wall. The diatom cells contain one nucleus located on the cytoplasmic bridge and chloroplasts, which are sparse and elongated in Pennales, and numerous and disc-shaped in Centrales. Nevertheless, it has been found that at least some of them can live in the dark thanks to their ability to uptake dissolved compounds such as glucose from the environment. The product of photosynthesis is fat deposited in the form of droplets in the plasma, and in large amounts also in the cell juice. Even these features indicate the need for a clear separation, even at the stage of reporting the results of both these systematic groups - diatoms and dinoflagellates. In the text, the species from these two groups are often described alternately, e.g. in Table 1, but also in drawings, e.g. in Fig. 3.
(2) According to the cited publication by Fukao et al. (2010) TEP secretion may be specific for particular diatom species - "The C. granii TEP production rate was highest in the growth phase, whereas those in E. zodiacus, R. setigera, and Skeletonema sp. Were highest in the stationary – decline chase ". This aspect was not addressed in the "Introduction", nor in the selection of the taxa studied. The "Introduction" also did not mention the influence of different ranges of temperature tolerance by diatom species, as noted by Chen et al. (2021).
Meanwhile, these aspects turned out to be one of the conclusions of the research: “In conclusion, the amount of TEP produced by phytoplankton may be associated with the species physiologies and growth patterns, so the relationship between TEP and the physiological" state of algae cells is worth researching. (Lines 313-315). I believe that the section "Introduction" should be at least partially changed and supplemented with the above-mentioned problems. In the current version it looks as if the most important finding was obtained randomly.
Other remarks:
Line 76: Fukao et al. (2010) studied four species of diatoms: Coscinodiscus granii, Eucampia zodiacus, Rhizosolenia setigera and Skeletonema sp. I do not see T. grainii among them.
Line 85-86: The average annual temperature certainly does not vary so widely.
Line 110: "... cell dimension were determined using a phytoplankton" ???
Line 111: Does Nitzschia closterium look like a cylinder?
Lines 122-123: Were TEC analyzes performed once for each species? The first sentence only says that the samples were collected "during the stationary phase".
Lines 134-136: The presented equation, and especially the given range of the factor, probably comes from some publication. Please quote her.
Line 168: Why did you change the color indicating the temperature in figure 1e?
Line 168: Why does the very similar growth pattern of P. micans and analogous C. affinis at 20°C and 24°C result in a significantly different specific growth rate?
Lines 172-174: The increase in TEP secretion by S. trichoidea with temperature was generally small, were these differences certainly statistically significant, or, like the differences in TEP concentration relative to Chl a, were insignificant?
Lines 186-187: Too many parentheses in the reported cell volumes.
Line 236: The drawing is not legible. I propose to introduce a separate scale for these four tested species, e.g. at the right and upper sides of the rectangle (their range of variation is much smaller than the physiological parameters tested).
Line 252-254: This sentence represents the results. They were not listed so precisely in the "Results" section.
Line 268-269: Should be: Lepidodinium chlorophorum.
Line 274: Should be: Thalassiosira pseudonana.
Round 2
Reviewer 1 Report
The authors adequately addressed my previous comments. Some minor issues remain with language and style, e.g. in section 4 it reads
"Temperature can promote the physiological parameters of POC, DOC, TOC, and cell volumes of dinoflagellates, indicating that their carbon contribution to the marine carbon pool will be promoted in higher water temperatures within their growth temperature ranges. This is similar to the results of Xiao's study, that is, within the same temperature range, diatoms and dinoflagellates respond differently to their physiological characteristic [63]."
What are physiological parameters of POC? "...within their growth temperature ranges" should probably read '...will be enhanced at higher water temperature' (the rest of this sentence is not needed); algae do not respond to their physiological characteristic(s) but respond to changes in environmental conditions by adjusting their physiology.
Once again, the manuscript should be checked by a native speaker with with a strong background in biology.
Reviewer 2 Report
see attached

Reviewer 3 Report
The authors have significantly improved the manuscript by positively considering my previous comments. I propose to accept the manuscript for printing.
Author Response
Thank you very much for your guidance.